# Prognostic accuracy of antenatal Doppler ultrasound for adverse perinatal outcomes in low-income and middle-income countries: a systematic review

Sam Ali [1,2] Simelina Heuving,[2] Michael G Kawooya,[1] Josaphat Byamugisha [3] Diederick E Grobbee,[2] Aris T Papageorghiou,[4] Kerstin Klipstein-Grobusch [2,5] Marcus J Rijken[2,6]

For numbered affiliations see end of article.

**Correspondence to**
Sam Ali;
alisambecker@gmail.com

## ABSTRACT

**Objectives** This systematic review examined available literature on the prognostic accuracy of Doppler ultrasound for adverse perinatal outcomes in low/middle-income countries (LMIC).

**Design** We searched PubMed, Embase, Cochrane Library and Scopus from inception to April 2020.

**Setting** Observational or interventional studies from LMICs.

**Participants** Singleton pregnancies of any risk profile.

**Interventions** Umbilical artery (UA), middle cerebral artery (MCA), cerebroplacental ratio (CPR), uterine artery (UtA), fetal descending aorta (FDA), ductus venosus, umbilical vein and inferior vena cava.

**Primary and secondary outcome measures** Perinatal death, stillbirth, neonatal death, expedited delivery for fetal distress, meconium-stained amniotic fluid, low birth weight, fetal growth restriction, admission to neonatal intensive care unit, neonatal acidosis, Apgar scores, preterm birth, fetal anaemia, respiratory distress syndrome, length of hospital stay, birth asphyxia and composite adverse perinatal outcomes (CAPO).

**Results** We identified 2825 records, and 30 (including 4977 women) from Africa (40.0%, n=12), Asia (56.7%, n=17) and South America (3.3%, n=01) were included. Many individual studies reported associations and promising predictive values of UA Doppler for various adverse perinatal outcomes mostly in high-risk pregnancies, and moderate to high predictive values of MCA, CPR and UtA Dopplers for CAPO. A few studies suggested that the MCA and FDA may be potent predictors of fetal anaemia. No randomised clinical trial (RCT) was found. Most studies were of suboptimal quality, poorly powered and characterised by wide variations in outcome classifications, the timing for the Doppler tests and study populations.

**Conclusion** Local evidence to guide how antenatal Doppler ultrasound should be used in LMIC is lacking. Well-designed studies, preferably RCTs, are required. Standardisation of practice and classification of perinatal outcomes across countries, following the international standards, is imperative.

## Strengths and limitations of this study

► This systematic review used the most optimal database combinations and snowballing technique with no time restrictions to identify the records.

► We comprehensively examined available literature on the prognostic accuracy of Doppler ultrasound for adverse pregnancy outcomes in low-income and middle-income countries.

► Although only English language articles were included, it is unlikely that high impact papers were not identified.

► Pooling and interpreting the data for wider clinical application was not possible due to the large heterogeneity across studies.

**PROSPERO registration number** CRD42019128546

## INTRODUCTION

Stillbirths remain a major global challenge,[1] with nearly three million cases reported annually.[2] The vast majority of the cases (98%) are contributed by low/middle-income countries (LMIC).[3] These deaths have profound effects on the families and communities involved, and strategies for reduction are of high societal importance. The risk of adverse perinatal outcomes is higher in compromised fetuses than in normally growing babies, and could be distinguishable using antenatal Doppler ultrasound.[4 5] Prenatal diagnosis of fetuses at risk provides a window for close monitoring and/or expedited delivery of well-developed babies with the prospect of improving survival and long-term well-being.[4]

The predictive performance of Doppler ultrasound for adverse perinatal outcomes has been demonstrated in primary studies, systematic reviews and meta-analysis from

high-income countries (HIC), guiding the development of HIC practice guidelines.[6] The use of HIC guidelines for clinical guidance in LMIC without local validation may be inappropriate given the differences in the prevalence of adverse pregnancy outcomes in the two settings. For instance, the stillbirth rates per 1000 total births (95% CI) is 3.4 (3.4 to 3.5) in HIC, compared to 25.5 (22.5 to 29.1) in Southern Asia and 28.7 (25.1 to 34.2) in sub-Saharan Africa.[2] Since the prevalence and severity of a disease influences the diagnostic or prognostic test performance, context-specific guidance is necessary.[7] However, there are still knowledge gaps about the predictive ability of antenatal Doppler for adverse pregnancy outcomes in LMIC.

This systematic review examined existing literature on the prognostic accuracy of Doppler ultrasound for adverse perinatal outcomes in LMIC. The implications for clinical utility of the available local evidence to guide practice in LMIC are highlighted.

## MATERIAL AND METHODS
### Protocol and registration
This systematic review protocol was registered in the PROSPERO database and reported following the Preferred Reporting Items for a Systematic Review and Meta-analysis of Diagnostic Test Accuracy Studies statement.[8]

### Eligibility criteria
We included observational (cohort or case–control) studies and randomised clinical trials (RCTs) from LMIC (as per the World Bank country classifications in the year 2020) reporting the prognostic value of Doppler ultrasound for adverse perinatal outcomes in singleton pregnancies of any risk profile. Doppler measurements of interest included umbilical artery (UA), middle cerebral artery (MCA), cerebroplacental ratio (CPR), uterine artery (UtA), fetal descending aorta (FDA), ductus venosus (DV), umbilical vein (UV) and inferior vena cava (IVC). Adverse perinatal outcomes (as defined in the included studies) were perinatal death, stillbirth, neonatal death, expedited delivery for fetal distress, meconium stained amniotic fluid, low birth weight, fetal growth restriction (FGR), admission to neonatal intensive care unit (NICU), neonatal acidosis, Apgar scores, preterm birth, fetal anaemia, respiratory distress syndrome (RDS), length of hospital stay, birth asphyxia and composite adverse perinatal outcomes (CAPO). Conference proceedings/posters that did not appear as full-text papers, case reports and review articles without original data were excluded.

### Information sources and search
We conducted a comprehensive literature search in PubMed (Medline), Embase, Cochrane Library and Scopus for articles published from inception to 7 April 2020. The search strategies (online supplemental

appendix S1) were developed with the support of a librarian at University Medical Center Utrecht. When applicable, predefined search (Title/Abstract) and MeSH/Emtree terms were used. No limits were applied to the searches.

### Study selection
The records retrieved from the databases were exported to Endnote to eliminate duplicates and then transferred to Rayyan for review and selection. Two reviewers (SA and SH) independently assessed all studies for inclusion based on title and abstract. Studies reporting any Doppler parameter and adverse pregnancy outcome of interest in the title or abstract were further retrieved in full text and assessed by the same two reviewers against full eligibility criteria. Disagreements were resolved by discussion or, if required, we consulted the third review author (MR).

### Data extraction
Using a pre-piloted data extraction sheet, two reviewers (SA and SH) independently extracted data on authors, study title, year of publication, aims of the study, study period, the number of women recruited, gestational age at Doppler ultrasound examination, method of pregnancy dating, pregnancy risk profile, blood vessels studied, pregnancy outcomes (as defined in the primary study) and key results. If any relevant information was missing, the corresponding authors were contacted once by email.

### Risk of bias assessment
Two raters (SA and SH) independently evaluated the risk of bias for each study using the quality in prognostic studies (QUIPS) tool.[9] The risk of bias domains included study population, attrition, prognostic factor measurement, outcome measurement, confounding and statistical analysis. All the domains were separately judged by two raters as having a low, moderate or high risk of bias. Any disagreement during this process was resolved by contacting the third rater (MR).

### Prognostic test accuracy measures
Doppler test prognostic performance measures, as reported in the selected studies, are presented in online supplemental table S1. These included diagnostic test accuracy measures such as sensitivity, specificity, positive predictive values (PPV) and negative predictive values (NPV); measures of association; proportions and correlations.

### Data synthesis and analysis
The results were narratively summarised. The large heterogeneity in the study populations, timing for Doppler tests, outcome definitions and prognostic performance measures in the included studies did not allow for a meta-analysis. If a study reported multiple Doppler indices, the most commonly used (pulsatility index) was selected.

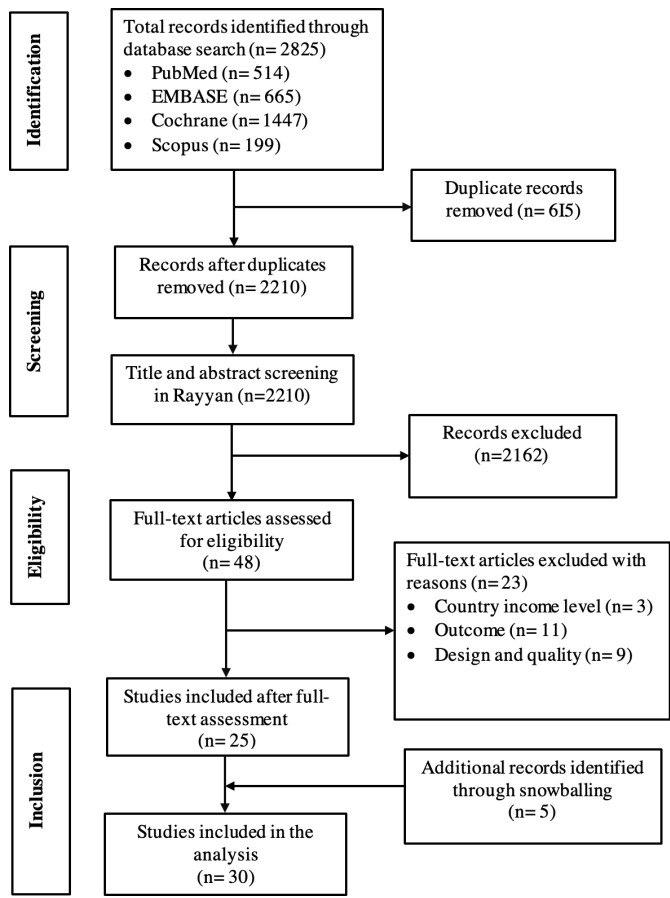

**Figure 1** Preferred Reporting Items for Systematic Reviews and Meta-Analyses flow diagram.

## Patient and public involvement

No patient was involved. The public was also not involved in the design, conduct and dissemination of this research.

## RESULTS

### Study selection

The 2825 records we identified through electronic searches were reduced to 2210 after the removal of duplicates, and 2162 were further excluded based on title and abstract screening, retaining 48 records. After full-text assessment for eligibility, 23 studies were excluded with reasons, and 25 remained (online supplemental appendix S2). Five additional records were identified through snowballing (figure 1). Thirty studies, involving a total count of 4977 women and a median (IQR) sample size of 100 (30–181) were included in the analysis (table 1).

### Study characteristics

The selected studies were from Africa (40.0%, n=12), Asia 17 (56.7%, n=17) and South America (3.3%, n=01). Twenty studies (67%) recruited high-risk pregnancies, six (16.7%) both high-risk and low-risk populations, while five (16.7%) studied the low-risk group (online supplemental appendix S3). Thirteen (43.3%) studies did not specify a method of pregnancy dating, 13 (43.3%) assessed gestational age using last menstrual period

(LMP) combined with ultrasound, 3 (10.0%) used ultrasound alone and 1 (3.3%) study used LMP. No RCTs were identified, and no study provided data on the UV and IVC Dopplers (table 1). The reasons for undertaking the Doppler research varied by individual studies and included the prediction of the risk of FGR, fetal anaemia, neonatal acidosis, among others (online supplemental appendix S3).

### Methodological quality of included studies

The results of the QUIPS assessment are provided in figure 2 and online supplemental appendix S4. Overall, the risk of bias was low in 15 (50%), moderate in 10 (33.3%) and high in 5 (16.7%) studies. In the study population domain, the risk of bias was low in 73.3%, moderate in 23.3% and high in 3.3% of the studies. Selective reporting remarkably resulted in a moderate to high risk of bias for analysis and reporting in 20 (66.7%) studies. We found a moderate to high risk of bias for outcome measurement in 17 (56.7%) studies, mostly due to inconsistencies in outcome classifications (online supplemental table S2).

### Prognostic accuracy of antenatal Doppler ultrasound for adverse perinatal outcomes

Twenty studies evaluated the UA,[10–29] and seven reported its predictive values for FGR. The PPV for FGR reported in the individual studies were between 77.40 and 88.5,[11 16 21 24] while the area under the receiver operating characteristic (AU ROC) curve was 0.63,[17] mostly in high-risk pregnancies. The NPV ranged from 55.4 to 95.65.[11 16 21 24] FGR was defined as birth weight or abdominal circumference below the 10th percentile in two studies,[11 17] ponderal index less than 10 in one study,[21] and was not defined in the remaining studies.[16 24 26] Increased flow impedance in the UA had PPV for composite adverse outcomes between 66.60 and 96.6 in high-risk pregnancies.[11 13 19 23] All studies provided individual components of the CAPO except only one.[11] Absent or reversed end-diastolic flow in the UA was associated with poor pregnancy outcomes (perinatal death, OR 9.8, 95% CI 2.1 to 46.4; CAPO: OR 2.4, 95% CI 1.1 to 5.0 and RDS: OR 8.4, 95% CI 2.3 to 30.5).[14 22 26]

The MCA was reported in 12 studies.[11–13 15 19 21 23 26 28 30–32] The PPV for fetal anaemia in Rhesus (Rh) isoimmunised pregnancies requiring transfusion were between 83.0 and 90.9 and the AU ROC curve was 0.7.[12 32] Fetal anaemia was consistently defined as haemoglobin (Hb)≤0.64 g/L in the two studies, though they recruited low numbers of women.[12 32] MCA Doppler had a sensitivity of 87.5%, PPV of 74.0% and AU ROC curve of 0.82 for neonatal acidosis.[30] The PPV for CAPO ranged from 80.0% to 100% in high-risk pregnancies,[11 13 19 23 31] but two studies did not provide details of the individual components of the CAPO.[11 31]

Nine studies reported the prognostic value of CPR.[11 13 15 19 20 23 26 33 34] CPR showed promising predictive value for adverse perinatal outcomes in unselected

**Table 1** Summary of studies included in the systematic review of current evidence on the prognostic value of Doppler ultrasound for predicting adverse pregnancy outcomes in LMIC

| Author | Country | Study period | Women | Weeks | Study design | Vessels | Abnormal Doppler thresholds |
|--------|---------|--------------|-------|-------|--------------|---------|------------------------------|
| Abdallah et al[10] | Egypt | 2015–2017 | 92 | ≥37 | Cohort | UA | UA (RI, PI and S/D ratio)>95th centile |
| Agbaje et al[17] | Nigeria | 2014–2015 | 120 | 26 | Cohort | UA | S/D ratio>95th percentile, RI>95th percentile and AREDF |
| Alanwar et al[33] | Egypt | 2017 | 100 | 30–40 | Cohort | CPR | CPR PI<1 or CPR PI<5th percentile |
| Allam et al[30] | Egypt | 2007–2010 | 30 | 36–41 | Cohort | MCA, DV | MCA S/D ratio<4.37, DV RI>0.29, or decrease in a-waves, v-waves and d- waves, or reversed flow in both a-waves and v-waves |
| Anshul et al[18] | India | 2005–2007 | 100 | ≥28 | Cohort | UA | S/D ratio≥3 or AREDF |
| Bano et al[11] | India | Not stated | 90 | 30–41 | Cohort | UA, MCA, CPR | MCA<2 SD; UA>2 SD or CPR PI<1.08 |
| Dhand et al[31] | India | 2005–2006 | 121 | 28–41 | Cohort | MCA | Not specified |
| Dorman et al[35] | Kenya | 1996–1997 | 854 | 24–31 | Cohort | UtA | Early diastolic notch or mean/ipsilateral UtA RI≥0.58 |
| Ebrashy et al[19] | Egypt | 2002–2003 | 80 | ≥28 | Case–control | UA, MCA, CPR | UA RI>0.72, MCA RI<0.69, CPR RI<1.0 |
| Geerts and Odendaal[20] | South Africa | Not stated | 113 | 24–34 | Cohort | UA, CPR, DV | UA PI>95th centile; UA/MCA>1; DV PI>95th centile |
| Khanduri et al[21] | India | 2009–2011 | 60 | 23–37 | Cohort | UA, MCA | UA PI>1.42 or UA RI>0.72, MCA PI<1.5, MCA RI<0.59 |
| Kumari et al[12] | India | 2015–2016 | 30 | | Cohort | UA, MCA, FDA | MCA PSV>1.50 MoM, FDA PSV delta>70.50. Not specified for UA |
| Lakhkar et al[13] | India | 2001–2002 | 58 | >30 | Cohort | UA, MCA, CPR, FDA | S/D ratio, RI or PI of UA>2 SD; MCA<5th centile; FDA>2 SD; CPR PI or S/D ratio<1.0 |
| Lakshmi et al[22] | India | 2007–2008 | 238 | <35 | Cohort | UA | Absent and/or reversed end-diastolic flow (AREDF) |
| Malik and Saxena[23] | India | 2010–2011 | 100 | 31–41 | Cohort | UA, MCA, CPR, UtA | Not specified |
| Masihi et al[34] | Iran | 2016–2017 | 181 | 38–40 | Cohort | CPR | CPR PI<1.94 |
| Mullick et al[24] | India | Not stated | 73 | 22–26, 30–32, >37 | Cohort | UA | S/D ratio≥4 (26 weeks), 3.5 (30–32 weeks) and 3 (37–40 weeks) |
| Nagar et al[25] | India | 2009–2011 | 500 | 26–30 | Cohort | UA, UtA | UA (S/D ratio or RI)>95th centile or AREDF. UtA S/D ratio>95th centile |
| Najam and Gupta[26] | India | Not stated | 150 | 28–40 | Cohort | UA, MCA, CPR | UA S/D ratio>2 SD, or AREDF, MCA SD ratio<5th percentile, MCA/UA SD ratio of <1.0 |
| Nouh and Shalaby[36] | Egypt | 2009–2011 | 80 | 8–12, 26 | Case-control | UtA | UtA PI>95th percentile, and/or unilateral or bilateral notch |
| Pares et al[32] | Brasil | 1997–2005 | 46 | 20–34 | Cohort | MCA, FDA | FDA-MV≥2SD MCA-PSV≥1.5 MoM |
| Pattinson et al[14] | South Africa | 1987–1989 | 53 | 16–28 | Cohort | UA, UtA | UA RI>95th centile UtA RI>0.58 |
| Pattinson et al[27] | South Africa | 1990 | 496 | 16–24 | Cohort | UA | UA RI>95th centile |
| Phupong et al[37] | Thailand | 2000–2001 | 322 | 22–28 | Cohort | UtA | Unilateral or bilateral early diastolic notch |
| Rani et al[15] | India | 2012–2014 | 223 | 30–36 | Cohort | UA, MCA, CPR | UA PI>1.03, UA RI>0.695; MCA PI<1.2, MCA RI<0.75; CPR PI<1.08 or CPR RI<1.05 |
| Rocca et al[16] | Egypt | Not stated | 113 | ≥28 | Cohort | UA | UA S/D ratio≥3 |

Continued

**Table 1** Continued

| Author | Country | Study period | Women | Weeks | Study design | Vessels | Abnormal Doppler thresholds |
|---|---|---|---|---|---|---|---|
| Verma and Gupta[38] | India | Not stated | 165 | 22–24 | Cohort | UtA | Bilateral diastolic notches or mean UtA PI>1.45 (UtA PI>95th centile) |
| Waa and Vinayak[28] | Kenya | 2007 | 100 | ≥28 | Cohort | MCA, UA | MCA RI<0.71 and UA>0.71 |
| Yelikar et al[29] | India | Not stated | 189 | >32 | Cohort | UA | UA S/D ratio>90th centile or AREDF |
| Zarean and Shabaninia[39] | Iran | 2015–2016 | 100 | 30–34 | Cohort | UtA | UtA PI>95th centile |

AREDF, absent and/or reversed end diastolic flow; CPR, cerebroplacental ratio; DV, ductus venosus; FDA, fetal descending aorta; LMP, last menstrual period; MCA, middle cerebral artery; MV, mean velocity; PI, pulsatility index; PSV, peak systolic velocity; RI, resistive index; S/D ratio, systolic diastolic ratio; UA, umbilical artery; UtA, uterine artery.

pregnancies in the third trimester. One study reported sensitivity 85.10, specificity 89.72, PPV 80.70 and NPV 92.30 for FGR.[26] Two studies found sensitivity between 80.90% and 90.91%, and specificity between 50.0% and 78.04% for emergency caesarean section for fetal distress though the tests had poor PPV.[26 34] Abnormal CPR had PPV for CAPO between 81.80% and 100% in high-risk pregnancies.[11 13 15 23]

Eight studies reported the prognostic value of UtA Doppler,[14 23 25 35–39] and two showed PPV of over 91.8% for CAPO in high-risk pregnancies.[23 36] The remaining studies had poor predictive values for adverse perinatal outcomes.

Three studies evaluated the prognostic accuracy of FDA Doppler.[12 13 32] The FDA sensitivity for fetal anaemia in Rh isoimmunised pregnancies ranged from 87.0% to 95.7% when used in isolation.[12 32] The sensitivity varied between 86.0% and 98.4% and PPV ranged from 86.0% to 100% when combined with the MCA.[12 32]

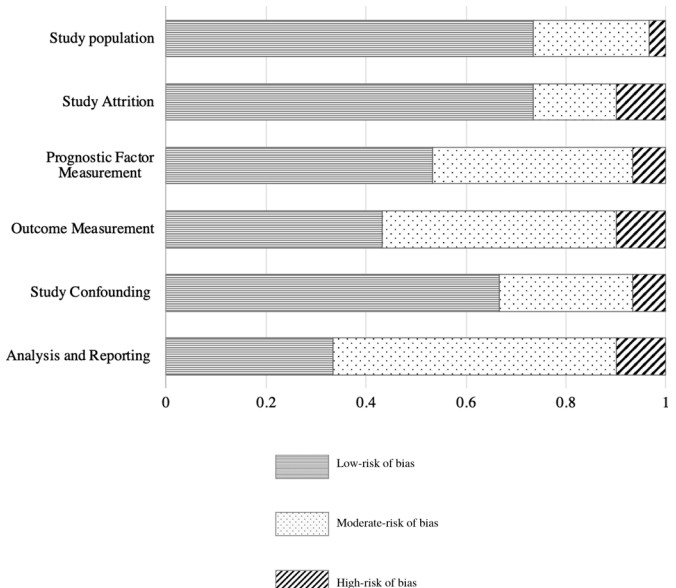

**Figure 2** Risk of bias assessment results of the 30 included studies.

The DV was sampled in two studies undertaken in high-risk pregnancies.[20 30] Abnormal DV had a sensitivity of 100, PPV of 72.0 and AU ROC curve of 0.88 for the prediction of neonatal acidosis, though this study included only 30 women between 36 and 41 weeks of gestation.[30] The second study found a borderline significance and positive predictive value of 92.0% for the prediction of CAPO at 24–34 weeks of gestation.[20]

## DISCUSSION
### Summary of findings
Many individual studies showed that abnormal UA Doppler was associated with poor perinatal outcomes, mostly in high-risk pregnancies, and that abnormal UA, MCA, CPR and UtA Dopplers had moderate to high predictive values for CAPO. A few studies suggested that abnormal MCA Doppler had high individual predictive value for fetal anaemia, but performed better when combined with the FDA. However, the majority of the available evidence was of suboptimal quality, based on a few poorly powered studies and had no RCTs. Further, wide variations in the populations studied, definitions of adverse perinatal outcomes and prognostic accuracy measures across studies was present. Thus, pooling and interpreting the evidence for wider clinical application was not possible.

### Implications for practice
Evidence from HIC suggests that adding Doppler studies into clinical diagnostic or prognostic rules improves pregnancy risk assessment,[6] and are increasingly becoming integrated into their pregnancy management guidelines.[4 6] The use of guidance based entirely on HIC data in daily practice in LMIC could be inappropriate considering the differences in the adverse pregnancy outcome rates in the two settings. The stillbirth rates in LMIC is approximately 10 times that of HIC,[2] a large variation likely to influence the predictive performance of diagnostic or prognostic tests.[7] Thus, a proper understanding of existing literature from LMIC is important. This paper

reports the findings of a systematic review of primary evidence on the prognostic value of antenatal Doppler ultrasound for adverse perinatal outcomes in LMIC.

Abnormal blood flow patterns in the UA had moderate to high predictive values for FGR and was associated with poor outcomes in high-risk pregnancies. Similarly, a recent Cochrane review of RCTs from HIC suggests that using UA Doppler in high-risk pregnancies could reduce perinatal deaths by 30% (risk ratio 0.71, 95% CI 0.52 to 0.98), and lead to fewer obstetric interventions.[40] Despite some similarities with our findings, the definitions of adverse outcomes, including FGR were inconsistent (or not even defined in many studies included in this review) with recommended international standards,[4 41] and with no clear distinction between early and late FGR. Scanty data from this review indicate that abnormal CPR, UA, MCA and UtA Doppler could be predictive of CAPO. However, in a previous systematic review from HIC, CPR had low predictive accuracy (pooled sensitivity: 57%, specificity: 77%, and summary positive likelihood ratio (LR): 2.5 and negative LR: 0.60) for CAPO in pregnancies with suspected FGR antenatally.[42] In another review, CPR was significantly better than UA and MCA Doppler in predicting CAPO (p<0.001) and emergency delivery for fetal distress in singleton pregnancies of all risk profiles,[43] but the primary studies reviewed had numerous methodological limitations.[43] Further, first-trimester UtA Doppler had very low sensitivity 25.8% (95% CI 15.5 to 39.7) for CAPO in a systematic review of 18 studies (involving 55 974 women).[44] More data from HIC indicate that MCA-PSV reliably predicts fetal anaemia in untransfused fetuses.[45] The area under the hierarchical summary ROC curve for moderate-severe anaemia in untransfused fetuses was 87%, pooled sensitivity 86% (95% CI 75% to 93%) and specificity 71% (95% CI 49% to 87%).[45] Similarly, in our study, MCA alone or when combined with FDA had high predictive values for fetal anaemia in Rh isoimmunised pregnancies, but this was based on only three studies. Overall, this review found that high-quality studies on the predictive accuracy of Doppler ultrasound for adverse perinatal outcomes in LMIC were scarce. The large heterogeneity across studies precluded a meta-analysis and between-study comparisons.

### Implications for research

Future studies need to specify the methods and timing for pregnancy dating. Accurate dating is crucial for timing the Doppler tests and interventions to expedite delivery in compromised fetuses. The interpretation and comparison of Doppler studies could be improved by using standard outcome definitions and completeness in reporting.[46] Most primary studies in this review studied the predictive ability of a single variable (Doppler test) for the outcome(s) of interest, without considering existing characteristics of clinical importance to estimate pregnancy risk. The predictive accuracies of new determinants need to be assessed individually and by multivariable analysis to facilitate the clinical applicability of the findings. The

clinical applicability of Doppler ultrasound also depends on the clinical judgement of the Doppler measurements and the feasibilities of local healthcare systems to interpret and respond to the results of the Doppler scan. Along the same line, our recently concluded prospective cohort study in a rural sub-Saharan African setting will soon highlight the prognostic value of Doppler ultrasound in the late third trimester and the feasibilities of integrating such advanced technologies into routine antenatal care in LMIC.

### Strengths and limitations

A strength of this systematic review is that it was conducted according to a registered protocol, using the most optimal database combinations and snowballing with no time restrictions. However, it is possible that some studies performed in low-resource settings may not have been indexed in the searched databases. Although we only included English language articles, it is unlikely that high impact papers were not identified. Further, this review primarily aimed to thoroughly examine the current evidence on the predictive value of Doppler ultrasound for adverse perinatal outcomes in LMIC using a meta-analysis. However, due to the inherent limitations in the included studies such as large heterogeneity in the study populations, inconsistencies in the definition of pregnancy outcomes, differences in the gestational age at the Doppler study and prognostic accuracy measures reported, we were only able to present our findings narratively. A future updated systematic review and meta-analysis of high-quality evidence is recommended.

### CONCLUSION

This review demonstrated that a scientific basis to provide evidence for how antenatal Doppler should be used in low/middle-income countries is lacking. Well-designed studies, preferably randomised controlled clinical trials, testing application models of antenatal Doppler while respecting the local conditions are needed. Moreover, local practice and classification of perinatal outcomes need to be standardised, utilising approaches consistent with international consensus.

**Author affiliations**
[1]Ernest Cook Ultrasound Research and Education Institute (ECUREI), Kampala, Uganda
[2]Julius Global Health, Julius Center for Health Sciences and Primary Care, University Medical Center Utrecht, Utrecht University, Utrecht, The Netherlands
[3]Department of Obstetrics and Gynecology, Makerere University College of Health Sciences, Kampala, Uganda
[4]Nuffield Department of Women's & Reproductive Health, John Radcliffe Hospital, University of Oxford, Oxford, UK
[5]Division of Epidemiology and Biostatistics, School of Public Health, Faculty of Health Sciences, University of the Witwatersrand, Johannesburg-Braamfontein, South Africa
[6]Department of Obstetrics and Gynecology, University Medical Center Utrecht, Utrecht, The Netherlands

**Acknowledgements** We thank the University Medical Center Utrecht librarians Paulien Wiersma and Felix Weijdema. We thank the team from Makerere University College of Health Sciences and ECUREI that supported us to undertake this review. We thank Fosca Tumushabe for the English language review.

**Contributors** SA is the guarantor in this review. SA, SH, KK-G and MR drafted the protocol and conducted the review. MK, JB, DEG and ATP critically reviewed the work for important intellectual content. All the authors approved the final manuscript.

**Funding** This research was supported by the UMC Utrecht Global Health Support Program (Ref: FM/ADB/D-18-015006), University Medical Center Utrecht, Utrecht, The Netherlands and Grand Challenges Canada (grant number: R-ST-POC-1808-17038).

**Competing interests** None declared.

**Patient consent for publication** Not applicable.

**Provenance and peer review** Not commissioned; externally peer reviewed.

**Data availability statement** No data are available. No additional data are available.

**ORCID iDs**
Sam Ali http://orcid.org/0000-0001-5459-2804
Josaphat Byamugisha http://orcid.org/0000-0002-3438-9662
Kerstin Klipstein-Grobusch http://orcid.org/0000-0002-5462-9889

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
