## [Reviewer comments · BMJ Open]

ARTICLE DETAILS

TITLE (PROVISIONAL)	Prognostic accuracy of antenatal Doppler ultrasound for adverse perinatal outcomes in low- and middle-income countries: a systematic review
AUTHORS	ALI, SAM; Heuving, Simelina; Kawooya, Michael; Byamugisha, Josaphat; Grobbee, Diederick; Papageorghiou, Aris; Klipstein-Grobusch, Kerstin; Rijken, Marcus

VERSION 1 – REVIEW

REVIEWER	Delius, Maria Ludwig-Maximilians-University , Obstetrics and Gynecology
REVIEW RETURNED	22-Mar-2021

GENERAL COMMENTS	dear authors, thank you for this systematic analysis. Though much is known on the topic, the authors are right in studying doppler in LMICs and especially further RCTs would be interesting. A very small mistake I guess -page 5 th emeconium stained liquor is probably meconium stained amniotic fluid.
--

REVIEWER	Al-Obaidly, Sawsan Hamad Medical Corporation, Obstetrics and Gynecology Department, Women's Wellness and Research Center
REVIEW RETURNED	24-Mar-2021

GENERAL COMMENTS	Thank you for your efforts in conducting this systematic review of the prognostic accuracy of antenatal Doppler for adverse perinatal outcome in the setting of low- and middle-income countries. Your search for relevant primary studies appears overall appropriate, however, these studies were with heterogenous methodology. It's unclear what were the indications of performing Doppler studies. In addition, FGR (if assuming this was the main indication for Doppler) was not defined in the majority of the studies. In table 1 some of the studies included pregnancies with gestational age before the age of viability! How is that relevant to this review? The term "High risk pregnancy" was consistently used in this review, however, this is a very broad terminology, again how many of these high-risk pregnancies were actually with or at risk of FGR?
--

REVIEWER	Sridharan, Kannan Seth GS Medical College and KEM Hospital, Department of Clinical Pharmacology
REVIEW RETURNED	21-Jul-2021

GENERAL COMMENTS	If meta-analysis is not done, it is erroneous to mention the OR with 95% CI in the abstract for various measures. It should only be descriptive.
--

REVIEWER	Bernasconi, Davide Università degli Studi di Milano-Bicocca, School of Medicine and Surgery
-----------------	--

REVIEW RETURNED	24-Jul-2021
-------------

GENERAL COMMENTS	This systematic review on the prognostic accuracy of Doppler ultrasound for adverse perinatal outcomes in low and middle-income countries is very well conducted. In particular, the assessment of the risk of bias in the studies included is thoroughly performed. Results are also clearly reported and the PRISMA guidelines are fully accomplished.
--

VERSION 1 – AUTHOR RESPONSE

Reviewer #1:

dear authors, thank you for this systematic analysis. Though much is known on the topic, the authors are right in studying doppler in LMICs and especially further RCTs would be interesting.

A very small mistake I guess - page 5 the meconium stained liquor is probably meconium stained amniotic fluid.

Thank you for your suggestion which we have taken into consideration. We have revised this in the abstract, [Line 40], and the main manuscript, [Line 110].

However, some authors do use the words “meconium stained liquor”. We leave this to the editor of this journal to decide whether liquor or amniotic fluid is the preferred wording.

Reviewer #2:

Thank you for your efforts in conducting this systematic review of the prognostic accuracy of antenatal Doppler for adverse perinatal outcome in the setting of low- and middle-income countries. Your search for relevant primary studies appears overall appropriate, however, these studies were with heterogenous methodology. It's unclear what were the indications of performing Doppler studies. Thank you for this valuable comment.

The reasons (aims) for undertaking the Doppler research varied by the individual studies. We added the primary aims of the selected studies to the online supplementary Appendix S3: The aims of the selected studies and risk profiles of the women recruited.

We also added a few lines to the main manuscript to further guide the reader, [Lines 176-178]. It reads like this:

The reasons for undertaking the Doppler research varied by individual studies and included the prediction of the risk of FGR, fetal anemia, neonatal acidosis, among others (online supplementary appendix S3).

In addition, FGR (if assuming this was the main indication for Doppler) was not defined in the majority of the studies.

Thank you for this observation and comment.

Yes, we agree with you and indicated in the main manuscript and the supplementary file (Table S2. Definitions of adverse perinatal outcomes reported in the selected studies) that FGR and other outcomes of interest were inconsistently (or not even) defined in many studies, [Lines, 240-242, and

259-262].

In addition, we had earlier indicated already in the manuscript that this review reported the outcomes as defined in the included studies, [Lines 108-109 and 135-136]. We further highlighted this as one of the hindrances for undertaking a meta-analysis, and narratively summarized the findings, [Lines 306-310].

In table 1 some of the studies included pregnancies with gestational age before the age of viability! How is that relevant to this review?
We thank you for this comment.

We aimed to examine and provide a good understanding of existing literature on the prognostic accuracy of Doppler ultrasound for adverse perinatal outcomes in LMIC, and discuss differences with available evidence from high-income countries and areas for further improvement in future research. The review is primarily intended to summarize available evidence to stimulate well-designed scientific studies in low- and middle-income countries.

We did not restrict on the gestation age at Doppler ultrasound as Doppler research has previously been conducted in Obstetric populations as early as the first trimester. For instance, the predictive value of the uterine artery for adverse pregnancy outcomes has been extensively investigated in high-income countries. [Source: Velauthar L, Plana MN, Kalidindi M, et al. First-trimester uterine artery Doppler and adverse pregnancy outcome: a meta-analysis involving 55 974 women. *Ultrasound Obstet Gynecol* 2014;43:500–7. doi:<https://doi.org/10.1002/uog.13275>].

The term “High risk pregnancy” was consistently used in this review, however, this is a very broad terminology, again how many of these high-risk pregnancies were actually with or at risk of FGR? Thank you for this comment.

The SMFM previously defined FGR as estimated fetal weight (EFW) below the 10th percentile. Recently, the SMFM revised the definition of FGR to EFW or abdominal circumference (AC) below the 10th percentile. On the other hand, the Delphi consensus criteria for FGR has been suggested as a better interim definition to cater for FGR even when fetal size is above the 10th percentile, although this is also largely based on expert opinions. [Sources:

Society for Maternal-Fetal Medicine (SMFM). Electronic address: pubs@smfm.org, Martins JG, Biggio JR, Abuhamad A. Society for Maternal-Fetal Medicine Consult Series #52: Diagnosis and management of fetal growth restriction: (Replaces Clinical Guideline Number 3, April 2012). *Am J Obstet Gynecol*. 2020 Oct;223(4):B2-B17. doi: 10.1016/j.ajog.2020.05.010. Epub 2020 May 12. PMID: 32407785.

Gordijn, S.J., Beune, I.M., Thilaganathan, B., Papageorgiou, A., Baschat, A.A., Baker, P.N., Silver, R.M., Wynia, K. and Ganzevoort, W. (2016), Consensus definition of fetal growth restriction: a Delphi procedure. *Ultrasound Obstet Gynecol*, 48: 333-339. <https://doi.org/10.1002/uog.15884>.

Lees, C.C., Stampalija, T., Baschat, A.A., da Silva Costa, F., Ferrazzi, E., Figueras, F., Hecher, K., Kingdom, J., Poon, L.C., Salomon, L.J. and Unterscheider, J. (2020), ISUOG Practice Guidelines: diagnosis and management of small-for-gestational-age fetus and fetal growth restriction. *Ultrasound Obstet Gynecol*, 56: 298-312. <https://doi.org/10.1002/uog.22134> ,

There are currently two recommended definitions of FGR, but none of the included studies adopted either of them because almost all of the selected studies, except about four (Abdallah, 2019; Alanwar, 2018; Kumari, 2019; Masihi, 2019), were conducted and published before the Delphi consensus and new SMFM criteria were introduced in 2016 and 2020 respectively.

Further, many studies that reported FGR as an outcome did not define it as indicated in the main

manuscript, [Lines, 240-242, and 259-262], and supplementary file (Table S2. Definitions of adverse perinatal outcomes reported in the selected studies). FGR was defined as birth weight or abdominal circumference below the 10th percentile in two studies [Agbaje, 2018; Bano, 2010] ponderal index less than 10 in one study [Khanduri et al., 2013] and was not defined in the remaining studies, [Lines, 194-196].

We reported the outcomes as defined in the included studies, [Lines 108-109 and 135-136]. Thus, pooling and interpreting the evidence for wider clinical application was not possible, [Lines 242-243]. Therefore, information on how many pregnancies were actually with or at risk of FGR was impossible to ascertain in the included studies.

As indicated in the manuscript, new and well-conducted studies in low- and middle-income studies are recommended.

Reviewer #3:

If meta-analysis is not done, it is erroneous to mention the OR with 95% CI in the abstract for various measures. It should only be descriptive.

Thank you for this critical observation.

We agree with you and have accordingly revised the abstract and main manuscript to only describe the key findings reported in the individual studies included in this review. The abstract results read like this:

Results We identified 2825 records, and 30 (including 4977 women) from Africa (40.0%, n= 12), Asia (56.7%, n= 17) and South America (3.3%, n= 01) were included. Many individual studies reported associations and promising predictive values of UA Doppler for various adverse perinatal outcomes mostly in high-risk pregnancies, and moderate to high predictive values of MCA, CPR and UtA Dopplers for composite adverse perinatal outcomes. A few studies suggested that the MCA and FDA may be potent predictors of fetal anemia. No randomized clinical trial was found. Most studies were of sub-optimal quality, poorly powered and characterized by wide variations in outcome classifications, the timing for the Doppler tests and study populations, [Lines 44-52].

The summary of key findings in the first paragraph of the discussion reads like this:

Many individual studies showed that abnormal UA Doppler was associated with poor perinatal outcomes, mostly in high-risk pregnancies, and that abnormal UA, MCA, CPR and UtA Dopplers had moderate to high predictive values for composite adverse perinatal outcomes. A few studies suggested that abnormal MCA Doppler had high individual predictive value for fetal anemia, but performed better when combined with the FDA. However, the majority of the available evidence was of sub-optimal quality, based on a few poorly powered studies and had no RCTs. Further, wide variations in the populations studied, definitions of adverse perinatal outcomes and prognostic accuracy measures across studies was present. Thus, pooling and interpreting the evidence for wider clinical application was not possible, [Lines 234-243].

Reviewer #4:

This systematic review on the prognostic accuracy of Doppler ultrasound for adverse perinatal outcomes in low and middle-income countries is very well conducted. In particular, the assessment of the risk of bias in the studies included is thoroughly performed. Results are also clearly reported and the PRISMA guidelines are fully accomplished.

Thank you very much for reviewing our paper.

VERSION 2 – REVIEW

REVIEWER	Al-Obaidly, Sawsan Hamad Medical Corporation, Obstetrics and Gynecology Department, Women's Wellness and Research Center
REVIEW RETURNED	19-Oct-2021

GENERAL COMMENTS	The previously made comments have been addressed by the authors
---

REVIEWER	Sridharan, Kannan Seth GS Medical College and KEM Hospital, Department of Clinical Pharmacology
REVIEW RETURNED	19-Oct-2021

GENERAL COMMENTS	Nil
-----